# A Flexible Thermocouple Film Sensor for Respiratory Monitoring

**DOI:** 10.3390/mi13111873

**Published:** 2022-10-31

**Authors:** Xiaodan Miao, Xiang Gao, Kaiming Su, Yahui Li, Zhuoqing Yang

**Affiliations:** 1School of Mechanical and Automotive Engineering, Shanghai University of Engineering Science, Shanghai 201620, China; 2Department of Otolaryngology-Head and Neck Surgery, Shanghai Jiao Tong University Affiliated Sixth People’s Hospital, Shanghai 200233, China; 3National Key Laboratory of Science and Technology on Micro/Nano Fabrication, Shanghai Jiao Tong University, Shanghai 200240, China

**Keywords:** respiratory monitoring, thermocouple, flow sensor, heat transfer

## Abstract

A novel flexible thermocouple film sensor on a polyimide substrate is proposed that is simple and flexible for monitoring the respiratory signal. Several thermocouples were connected in series and patterned on the polyimide substrate, and each one is formed by copper and a constant line connected to each other at two nodes. The respiratory signal was measured by the output voltage, which resulted from the temperature difference between the hot and cold junctions. The sensors were fabricated with surface-microfabrication technology with three sputtering steps. The measurement results showed that the peak voltage decreased by 90% in the case of apnea compared with normal breathing. The sensor has potential application for wearable detection of sleep apnea hypopnea syndrome (OSAHS).

## 1. Introduction

Obstructive sleep apnea hypopnea syndrome is a serious, potentially fatal sleep respiratory disease that seriously affects the physical and mental health of patients [1]. The respiratory signal is the most important parameter of OSAHS detection. The respiratory sensor can detect sudden infant death syndrome or record a patient’s physiological status for sleep studies and sports training [2]. The overall prevalence rate of OSAHS is 9% to 38% in the general adult population, and is much higher in the elderly population [3]. In clinical practice, polysomnography is the standard diagnostic method for OSAHS [4].

According to the classification criteria of the American Academy of Sleep Medicine (AASM), the detection and diagnosis of OSAHS can be divided into four levels: Level 1: standard polysomnography, Level 2: comprehensive portable polysomnography, Level 3: modified portable sleep-apnea testing, and Level 4: continuous single or dual bio-parameter recording. With the expansion of the population and progress of technology, some new classifications were proposed [5,6].

In traditional sleep-monitoring methods, polysomnography is a common device, which is mainly used in hospitals and sleep laboratories. Polysomnography can record a variety of physiological signals of the human body in the process of sleep all night, including electrocardiogram (ECG), electroencephalogram (EEG), electrooculogram (EOG), electromyogram (EMG), respiratory signal, oxygen saturation, and sleep posture [7]. However, due to its high cost and great mental pressure on patients, studies on portable sleep respiration-monitoring devices have become more popular in recent years [8].

Various sensors and sensing methods have been developed to measure respiratory rate and/or lung capacity, including transthoracic impedance, blood O_2_/CO_2_ concentration, and breathing airflow [9,10,11,12,13,14,15,16,17,18,19,20,21,22,23,24,25,26]. Breathing air flow is typically detected by sensing pressure or temperature, and the adopted sensors may be resistive, thermoelectrical, pyroelectrical, or piezoelectric [10,11,12,13,14,15,16,17,18,19,20,21,22,23,24,25,26]. Chen designed skin-like hybrid integrated circuits (SHICs) with stretchable sensors that could capture the temperature change of the inhaled and exhaled air. By integrating the flexible devices on human faces, comfort can be enhanced [9]. Jiang proposed a portable sleep respiration-monitoring system including three sensors that could monitor airflow, body posture, and oxygen saturation at the same time. The combination of the three signals could improve the accuracy of diagnosis of OSAHS [10]. Dalola reported a micromachined smart system for flow measurement based on a silicon substrate, consisting of four germanium thermistors embedded in a thin membrane and connected to form a Wheatstone bridge supplied with a constant DC current. It can measure the velocity and transport rate at the same time, which combines calorimetric and hot-wire transduction principles with lower power consumption and thermal loss [22]. Wei et al. represented a novel CMOS process-compatible MEMS sensor for monitoring respiration. This resistive flow sensor was manufactured by the TSMC 0.35 m CMOS/MEMS mixed-signal 2P4M polycide process. The sensor was demonstrated to be sensitive enough to detect the respiratory flow rate, and the relationship between flow rate and sensed voltage was linear [23].

Compared with the work in Refs. [22,23], in which the air-flow sensor was fabricated based on silicon substrate and supplied with a constant DC current, this paper presents a thermocouple thin-film flow sensor that was fabricated based on a polymide substrate with three sputtering steps based on the thermal electromotive force without an extra power supply. As a result, it is more flexible for a wearable diagnosis system outside the hospital with lower cost. It consists of four micromachined thermocouples on a polyimide substrate with a translation circuit. Its working principle is based on the combination of the Seebeck effect and the heat-transfer theorem. The thermocouple is formed by two different metal lines connected to each other at two nodes. One node serves as a hot junction, and its surrounding temperature changes with the respiratory process. The free node serves as cold junction, and its surrounding temperature keeps constant. As a result, the thermal potential is generated from the temperature difference between the hot and cold junctions. The respiratory signal is measured by the output voltage, which is converted from the thermal potential. It was fabricated by surface microfabrication technology and the testing results show that the peak voltage decreased by 90% in the case of apnea compared with normal breathing. The sensor has potential for application to the detection of sleep apnea hypopnea syndrome (OSAHS).

## 2. Physical Modeling

### 2.1. Physical Model

The flexible thermocouple sensor consists of four thermocouples, as shown in Figure 1. Each thermocouple is formed by two different metal lines consisting of copper and constantan, which are connected to each other at two nodes. One node serves as a hot junction, and its surrounding temperature changes with the respiratory process. The free node serves as a cold junction, and its surrounding temperature keeps constant. As a result, the thermal potential is generated from the temperature difference between the hot and cold junctions. Under the action of respiration, the temperature of the hot junction increases or decrease and the cold junction remains unchanged, resulting in a temperature difference and thermal electromotive force. Finally, the electromotive-force signal is converted into a respiratory signal.

### 2.2. Mathematical Model

The sensor works based on the Seebeck effect, which means that if two different kinds of metals are connected at two nodes with different temperatures, an electric current is generated [17]. The Seebeck effect can be expressed as follows:(1)$$E_{AB}\left(T,T_{0}\right)=\frac{k}{e}\left(T-T_{0}\right)\ln\frac{n_{A}}{n_{B}}+\int_{T_{0}}^{T}\left(\sigma_{A}-\sigma_{B}\right)dT$$
where *E_AB_* is the thermal electromotive force of the two materials and *S_AB_* is the Seebeck coefficient. *K* is Boltzmann’s constant, *E* is the electron charge, *T* is the temperature of the hot junction, and *T_0_* is the temperature of the cold junction. Using *S_AB_* as the Seebeck coefficient of the two materials, *E_AB_* can be expressed as the integral of the Seebeck coefficient over the temperature difference per the following:(2)$$E_{AB}=\int_{T_{0}}^{T}S_{AB}\left(T\right)dT$$

Based on different materials, the thermocouples can be divided into many types. The thermocouple type used in this paper was the *T* type, which is composed of copper and constantan.

The Seebeck coefficient of the thermocouple is related to the temperature of the material and the size effect. According to the research by Yang et al. [21], the Seebeck coefficient of a *T*-type thermocouple is 43.98~46.47 μV/°C when the thickness of thermocouple is from 0.5 to 2 μm.

The change in respiratory airflow is approximated as a sine function. It is known that the tidal volume of normal people in a sleep state is about 0.6 L [18,19] and a breathing cycle is 3 s. The change in gas volume (*Q*) of human lungs with time (*t*) can be expressed as following:(3)$$Q(t)=0.3\sin(\frac{2\pi t}{3}-\frac{\pi }{2})+0.3$$
*A* (*A* = 100.48 mm^2^) is used to represent the area of the nasal entrance of normal people, and the air velocity *V* at the entrance can be expressed as follows:(4)$$V(t)=\frac{dQ(t)}{dt}\times \frac{1}{A}=6.25\cos(\frac{2\pi t}{3}-\frac{\pi }{2})$$

The viscosity (*ν*) and Planck number (Pr) can be obtained. The total length (l) of the heat-transfer part is 5 mm. The Reynolds number (Re*_l_*) can be calculated with Equation (5). It is a laminated flow, since Re_l_ is less than Re*_c_*.
(5)Rel=V(t)lν

The average convective heat-transfer coefficient when the laminar flow flows along a flat plate can be expressed in Formula (6) as follows:(6)h=0.664λlRel1/2Pr1/3

The calculation results of each parameter are shown in Figure 2 and are used as the boundary condition for the finite-element simulation as follows.

## 3. Simulation

### 3.1. Simulation of the Distribution of the Polyimide Substrate

The important parameters of the sensors were simulated, such as the polyimide film thickness and the distribution of the thermocouples under different thermal conditions. Firstly, the static-temperature distribution of the polyimide film was simulated in case of no breathing. The skin temperature was set to 36 °C and the ambient temperature was set to 20 °C. The natural convection heat-transfer coefficient was set to 5 W/(m^2^·°C), which decided the heat transfer and the temperature-equilibrium point in steady state. The steady-state results obtained by COMSOL simulation are shown in Figure 3. The temperature on the main part of the polyimide was 36 °C, nearly the temperature in the nostril because of the heat transfer. In comparison, two isosceles-square parts were kept at 20 °C, which were suspended and separated from the skin and kept constant with the air.

Secondly, the temperature on the center position with different thicknesses of the polyimide film was simulated. The results indicate that the response speed of the temperature decreased slightly with the increase in the thickness, as shown in Figure 4. However, the central steady-state temperature of polyimide film was kept at a stable state. As a result, the thickness of the polyimide film had less influence on the temperature variation of the polyimide film.

Thirdly, the distribution of the cold and hot junctions was simulated. The heat-transfer efficiency of the polyimide film along the thickness was larger because its thickness was far less than the length and width. As a result, the center temperature of the film increased with the skin temperature. As shown in Figure 5, two isosceles-square parts of the thin film were less effected by the skin because they were suspended and separated from the skin in the nostril. The cold conjunctions were located on the two isosceles-square parts to keep them at a constant temperature.

The signal of a thermocouple is small, and a more stable and accurate temperature signal can be acquired by connecting several groups of thermocouples in a series, as shown in Figure 6. As shown in Figure 6a, the cold junction was located on the center part of the polyimide film, which was affected by the skin. In order to generate the temperature difference between the cold and hot junctions, the design of the thermocouple was optimized as shown in Figure 6b, where the cold junction was led out with extension wire to avoid the influence of the skin.

### 3.2. Simulation of the Single Thermalcouple

The thermoelectric coupling-effect simulation of a single thermocouple was carried out to study its thermal electromotive force. Copper and constantan were selected as the materials, and the Seebeck coefficient S_AB_ was 43 μV/°C. It was assumed that the ambient temperature was 20 °C and the respiratory temperature was 36 °C. Therefore, the temperature difference was 16 °C. Figure 7 shows that the thermal electromotive force was $7\times 10^{-4}$ V. According to Formula (2) mentioned above, the thermal electromotive force was $6.88\times 10^{-4}$ V. The results of the theoretical simulation and finite-element simulation tended to be consistent.

The transient simulation was carried out to study the influence of different thicknesses on the response speed of the sensor. On the premise of a fixed Seebeck coefficient, the relationship between the temperature difference and thermal electromotive force is linear. Therefore, the results can be expressed in terms of the temperature of the hot junction. The temperature setting was the same as before and the convective heat-transfer coefficient was set as 100 W/(m^2^·°C). In Figure 8, when the temperature reached 90% of the maximum temperature, the corresponding time was defined as the response time, as shown in Table 1.

With the decrease in the thermocouple thickness, the response speed increases. However, the smaller the thickness is, the more difficult the fabrication process. At the same time, the resistance value also increases, which would affect the signal acquisition.

### 3.3. Simulation of the Sensor under the Working Conditions

The polyimide film, the thermocouple, and the respiratory signal were combined to simulate the temperature variation of the sensor during breathing. The actual ambient temperature at 27 °C was used as a simulation parameter for comparison with the results of the measurement in the environment. In order to verify the ability of the sensor to detect respiration, the respiratory intensity was changed for simulation. If the respiratory intensity decreases by 30%, it can be considered hypopnea, and more than 50% is very serious hypopnea [3,4,5,6].

The convective heat-transfer coefficient was used as the simulation condition, and a parametric sweeping was carried out to study the temperature of the hot-junction change under the condition of hypopnea. As shown in Figure 9, the temperature curves of hypopnea and normal respiration were significantly different. The average temperature difference of the hot junction in multiple respiratory cycles under different respiratory flow is shown in Table 2. The decrease range of the average temperature difference relative to 100% respiration intensity can be used as a criterion to estimate whether hypopnea occurs.

As shown in Figure 9, the maximum temperature of the hot junction of the thermocouple did not exceed 32.5 °C, but human body temperature was at least 36 °C. It was assumed that the PI membrane absorbed part of the heat. In order to verify this hypothesis, a simulation was carried out by changing the PI film thickness while keeping the respiratory flow unchanged. The simulation results are shown in Figure 10. The average temperature difference under different film thicknesses is shown in Table 3.

The results show that the thickness of the polyimide film had a great influence on the temperature difference due to the heat absorption by the polyimide film. The larger the thickness of the polyimide film, the smaller the average temperature difference of the thermocouple hot junction in multiple breathing cycles. It can be concluded that an increase in polyimide film thickness leads to a decline in respiratory-monitoring performance of the sensor. Therefore, reducing the thickness can effectively improve the quality of the signal.

## 4. Microfabrication

In the design and simulation mentioned above, the key parameters were optimized, such as the distribution of the thermocouple, the thickness of the polyimide, and the linewidth of the thermocouple. The flexible thermocouple sensor was fabricated with microfabrication technology, which was mainly composed of sputtering technology. The fabrication process was as follows:

The glass wafer and the polyimide film (PI film) were ultrasonic cleaned in deionized water and finally dried in an oven at 60 °C for 30 min. The thickness of the PI film was 100 μm.

(1)The cleaned PI film was attached to the glass substrate and spin-coated with AZ-4620 positive photoresist with a thickness of 10 μm. Then the glass substrate was heat treated.(2)The substrate was exposed for 30 s and then developed for 15 s to form the mold of the bottom layer for sputtering.(3)Then, the chromium seed layer was sputtered at 300 W for 180 s and the copper layer at 300 W for 480 s. The process of sputtering copper was repeated five times.(4)The photoresist was removed by ultrasonic cleaning in acetone solution for 30 min.

After removing the photoresist, the mask was replaced and the lithography steps were repeated. After sputtering the chromium and constantan layers, the photoresist was removed at last. It should be noted that in order to ensure the adhesion of the two metal layers, the area of the two masks layer need to be partially overlapping. The manufacturing process is shown in Figure 11. The thickness of the sputtering metal measured by step profiler was 500 to 600 nm.

## 5. Experiment and Results

### 5.1. Microscopic Morphology of the Sensor

The samples were analyzed by ultra-high-resolution FE-SEM, as shown in Figure 12. It can be observed that the upper part of the wire was constantan, which contains both nickel and copper. The lower part was copper wire. In addition, the overlapping of copper and constantan could be clearly observed by magnifying the junction of the two materials.

### 5.2. Sleep Apnea Measurment System and Results

The measurement system was established as shown in Figure 13. The output of the sensor was connected to the oscillographs and an amplifier with a power supply of 1.5 V × 4. In the measurement system, the thermocouple signal was weak and needed to be amplified. A kind of low-power, high-precision amplifier AD620 was applied, which could set the magnification from 1 to 1000 times with only one external resistor. The maximum input offset voltage was 50 μv. The circuit diagram and the print circuit board (PCB) are shown in Figure 14. The ambient temperature was 27 °C, and the multiple of the amplifier was set to 100 times.

Then, the sensor was attached to the bottom of the nose, with the input part inside the nose and the output part outside the nose, as shown in Figure 15. The different respiratory signal of the sensor was tested by the oscilloscope when the normal and abnormal respirations were modified. The measurement results were obtained as shown in Figure 15, Figure 16 and Figure 17. As shown in Figure 15, the thermal electromotive-force curve changed periodically with respiration and the peak voltage was 22.4 mv. The temperature of the thermocouple increased in the expiratory phase and decreased in the inspiratory phase. One of the breathing cycles was taken out and converted to the actual temperature, and the maximum temperature was 32.23 °C. Compared with the simulation results, the maximum temperature difference between the test and simulation results was 5.38 °C, as shown in Figure 16 and Figure 17, which indicates that the tested results and the simulation results tended to be consistent.

A student attached the flexible sensor on the face beneath the nostril and imitated the breathing of sleep apnea, which can be diagnosed by recognizing a significant decrease in the thermoelectromotive force, as shown in Figure 18. Under normal breathing conditions, there were peaks and troughs in the curve. The peaks indicate the end of exhalation and the troughs indicate the end of inspiration. Because of the noise in the actual detection, there was also a signal in apnea. However, the peak value of apnea decreased more than 90% compared with that of normal breathing, which indicates the occurrence of apnea—at least, very serious apnea.

## 6. Conclusions

This paper presents a flexible thermocouple film sensor for respiratory monitoring, which has potential as an OSAHS-diagnosis system. Since it was fabricated on a flexible polyimide substrate, the sensor could be attached to the skin softly and be portable outside the hospital in the monitoring process. The sensor works based on the electromotive force from the temperature difference between the hot and cold junctions of the polyimide-film thermocouple, based on the combination of the Seebeck effect and the heat-transfer theorem.

In order to acquire the temperature difference between the cold and hot junctions, the cold junction was led out to keep at a constant temperature, whereas the average temperature of the hot junction varied with the breathing air flow. The influence of the thickness of the thermocouple and the polyimide film on the sensors was simulated and optimized. By reducing the thickness of the thermocouple, the response speed could be effectively improved as well as the defective rate, whereas the difficulty of signal measurement was increased. Simulation results also show that reducing the thickness of the polyimide film could effectively improve the signal quality. However, the polyimide film absorbed part of the heat, so the sensor could not reach the maximum temperature.

The sensor was tested in the case of apnea and the peak voltage decreased by 90%, which is in agreement with the simulation results. The results indicate that the flexible thermocouple film sensor could be used in respiratory monitoring for OSAHS detection. In future studies, the stability and wearability, reliability, and intellectualization can be improved and optimized.

## Figures and Tables

**Figure 1 micromachines-13-01873-f001:**
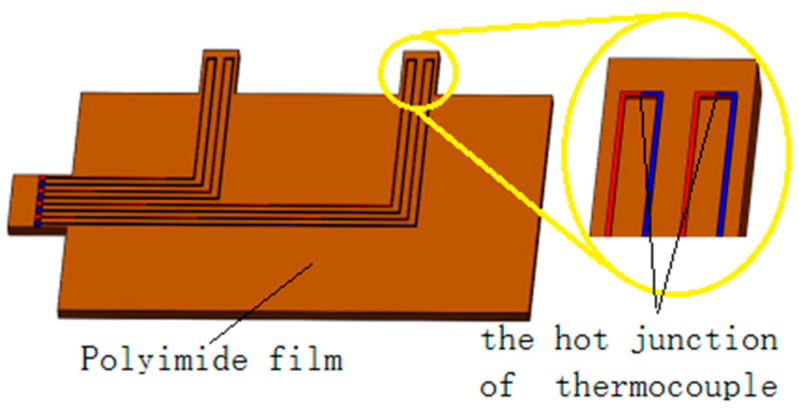
Physical model of the flexible thermocouple film sensor.

**Figure 2 micromachines-13-01873-f002:**
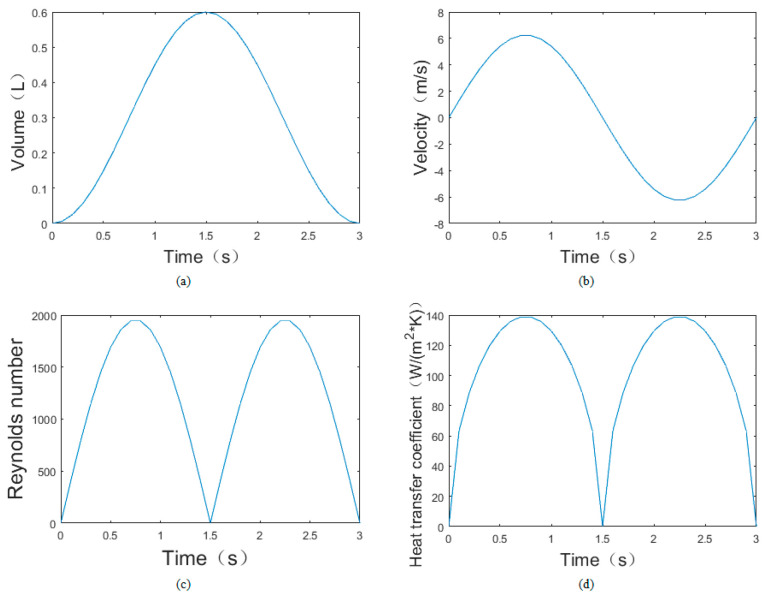
(**a**) The lung gas-volume variations with time; (**b**) the nasal airflow-velocity variations with time; (**c**) the Reynolds-number variations with time; (**d**) the convective heat-transfer coefficient variations with time.

**Figure 3 micromachines-13-01873-f003:**
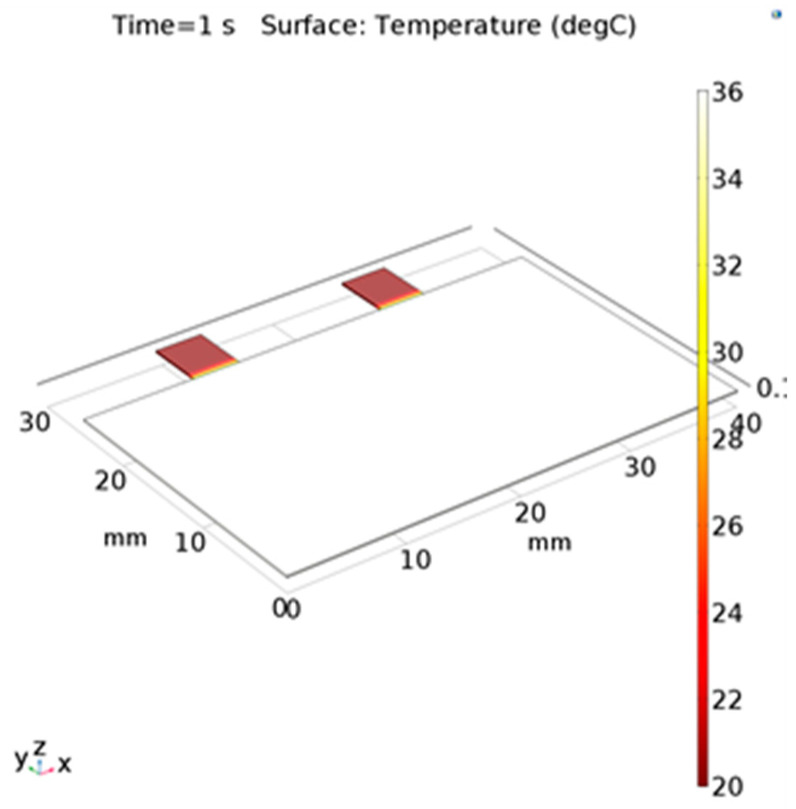
Steady-state simulation results of the polyimide film.

**Figure 4 micromachines-13-01873-f004:**
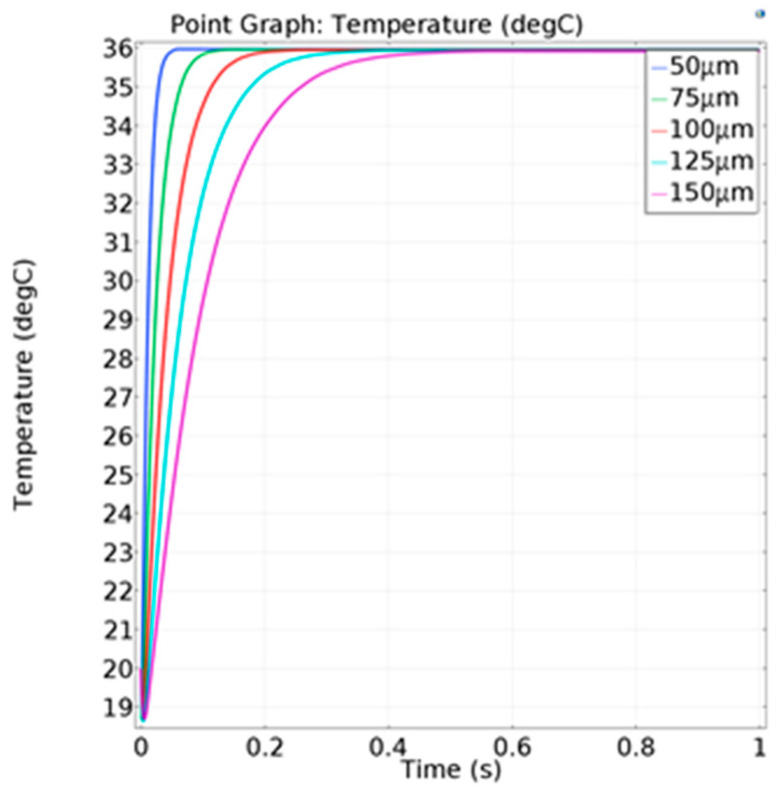
The temperature variation of the film center under different film thicknesses.

**Figure 5 micromachines-13-01873-f005:**
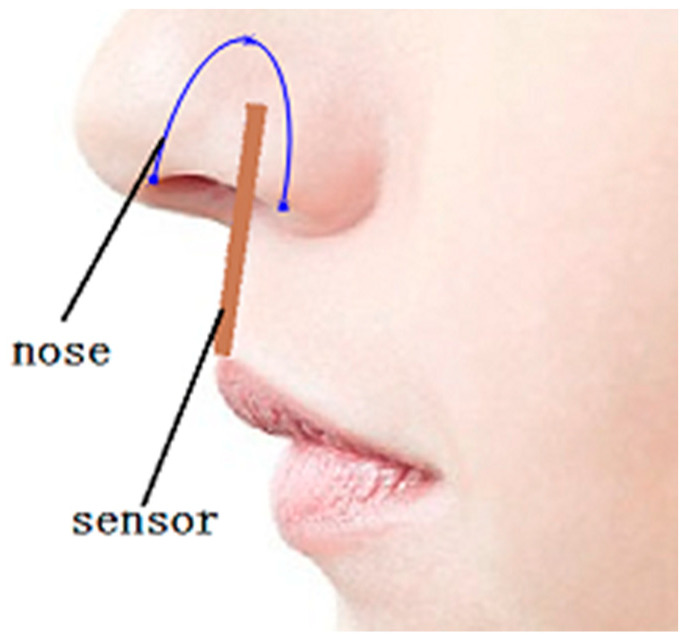
Schematic diagram of sensor position.

**Figure 6 micromachines-13-01873-f006:**
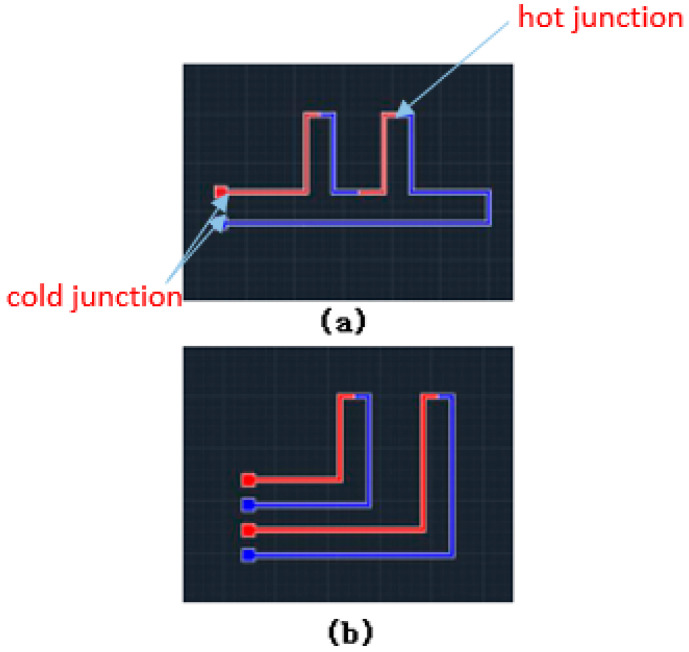
Two structures of the thermocouple. (**a**) The cold junction is located on the center part of the polyimide film; (**b**) the cold junction was led out with extension wire to avoid influence from the skin.

**Figure 7 micromachines-13-01873-f007:**
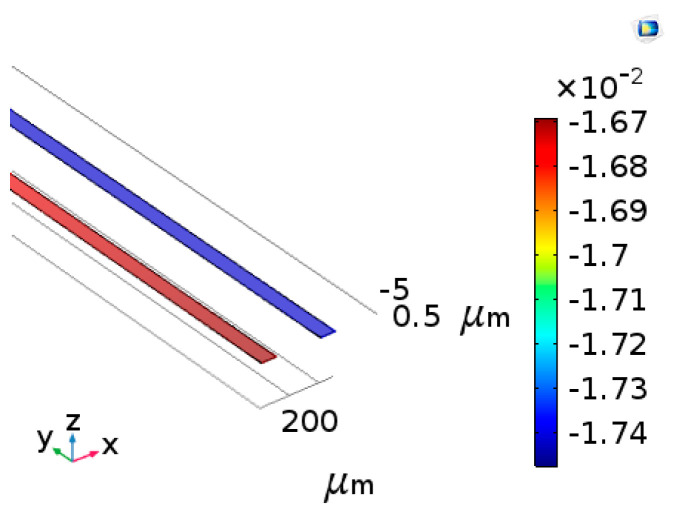
The simulation results of the cold junction of the thermocouple.

**Figure 8 micromachines-13-01873-f008:**
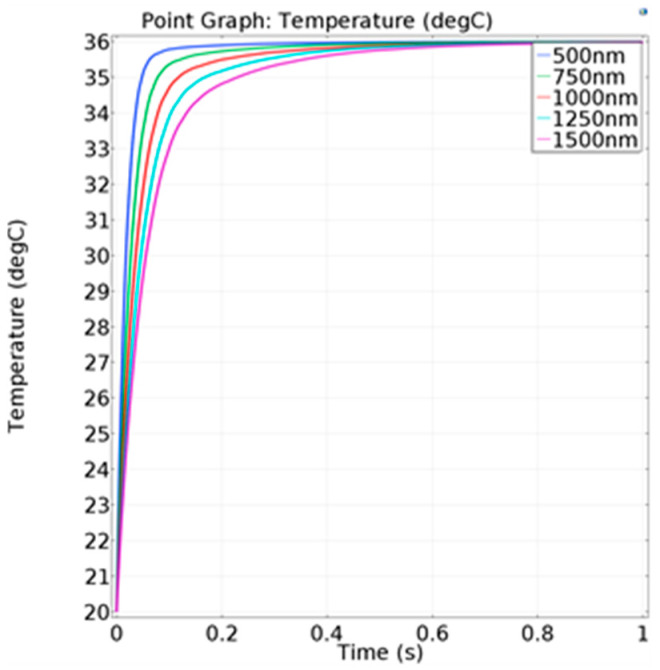
Temperature change of the hot junction at different thermocouple thicknesses.

**Figure 9 micromachines-13-01873-f009:**
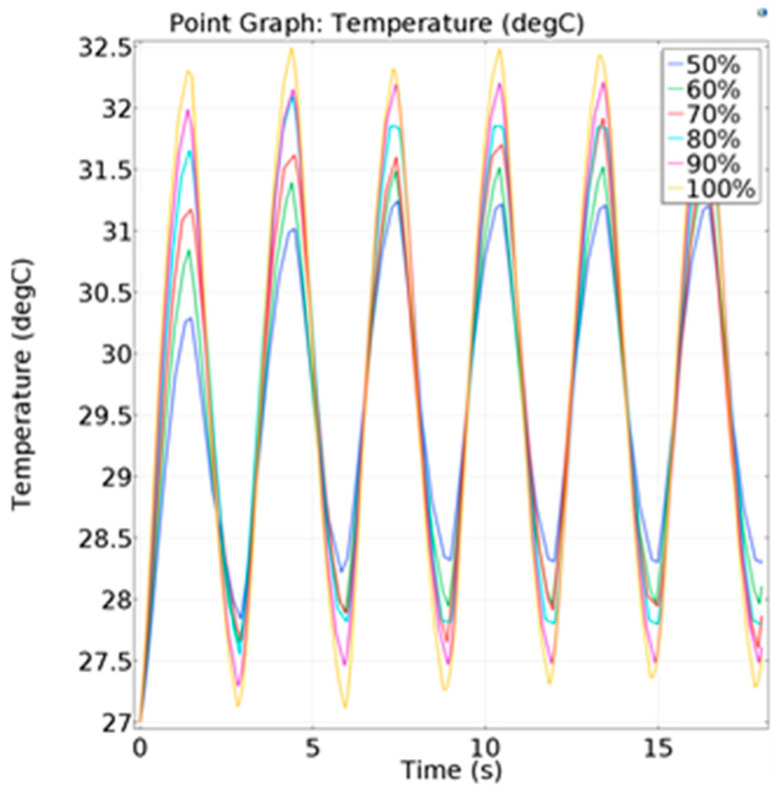
Temperature change of the hot junction under different respiratory flows.

**Figure 10 micromachines-13-01873-f010:**
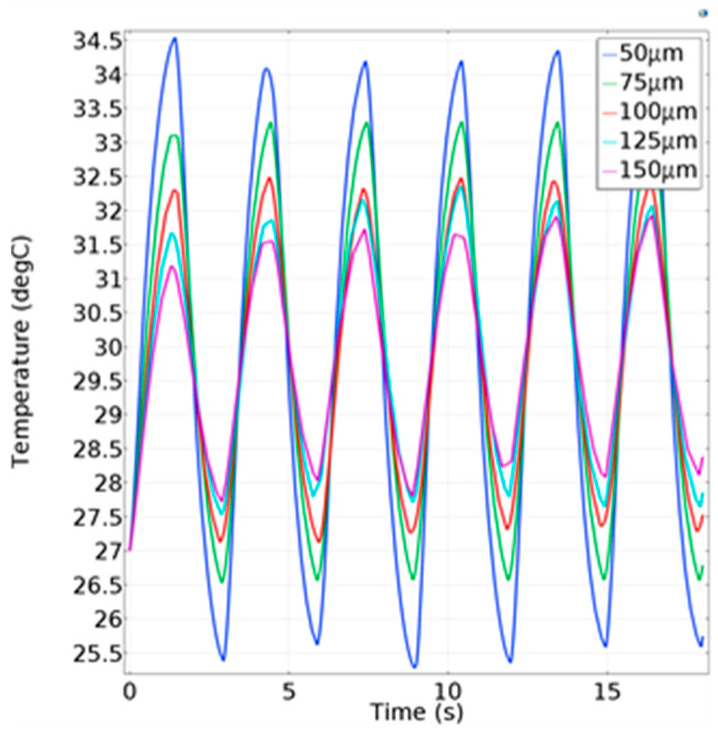
Temperature change of the hot junction under different thin-film thicknesses.

**Figure 11 micromachines-13-01873-f011:**
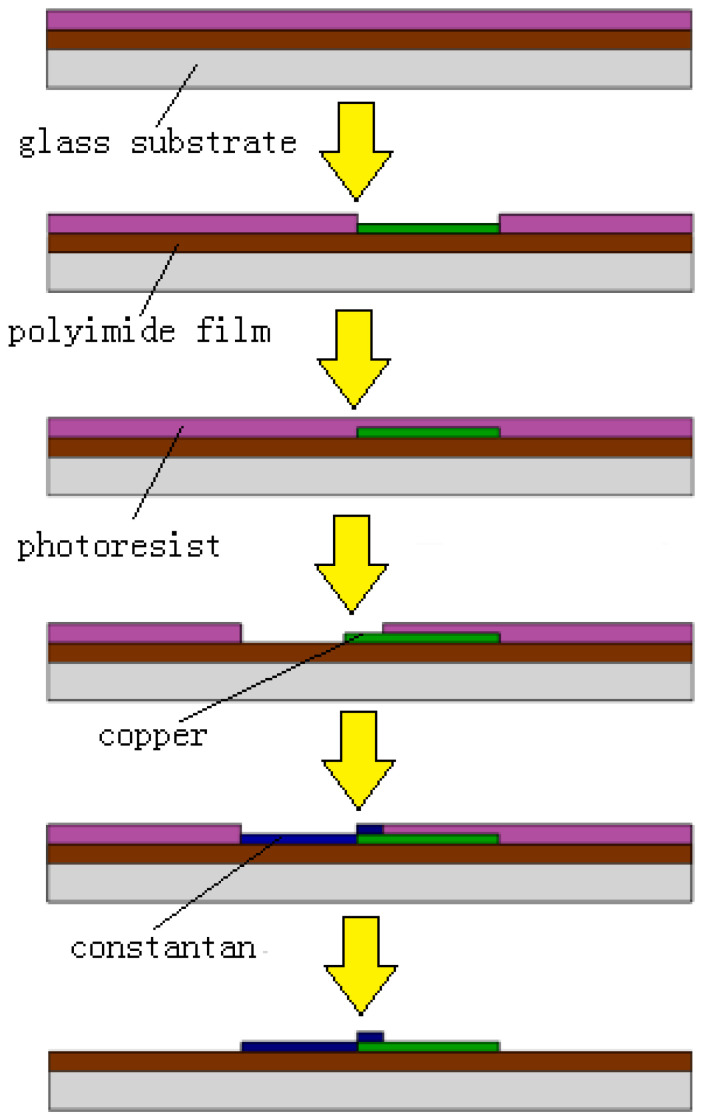
Fabrication-process diagram of the sensor.

**Figure 12 micromachines-13-01873-f012:**
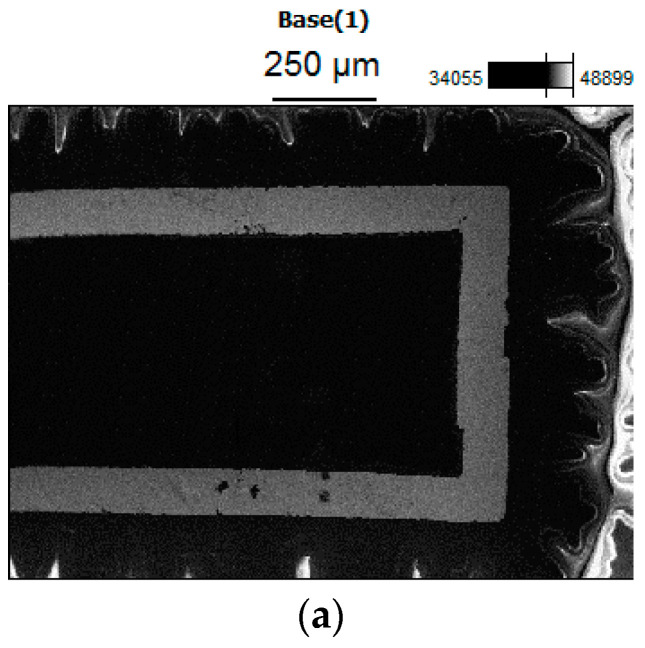
SEM images of thin-film thermocouple. (**a**) Connection point of thermocouple. (**b**) Distribution of Ni. (**c**) Distribution of Cu.

**Figure 13 micromachines-13-01873-f013:**
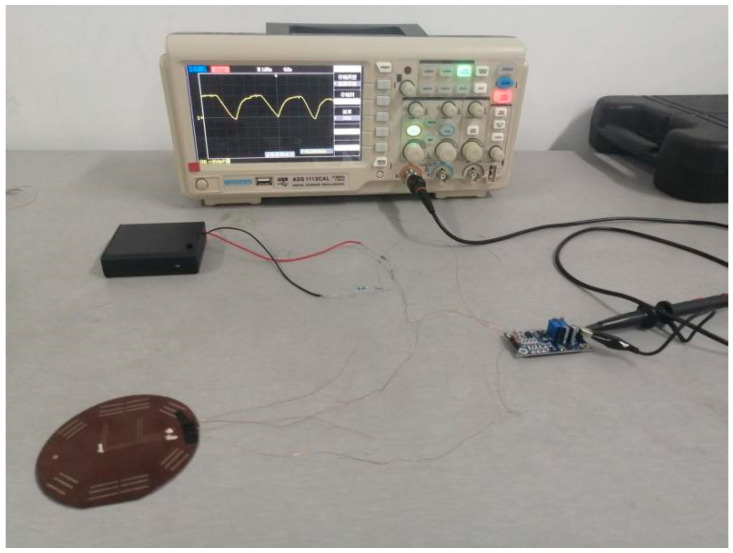
The measurement system of the flexible thermocouple film sensor.

**Figure 14 micromachines-13-01873-f014:**
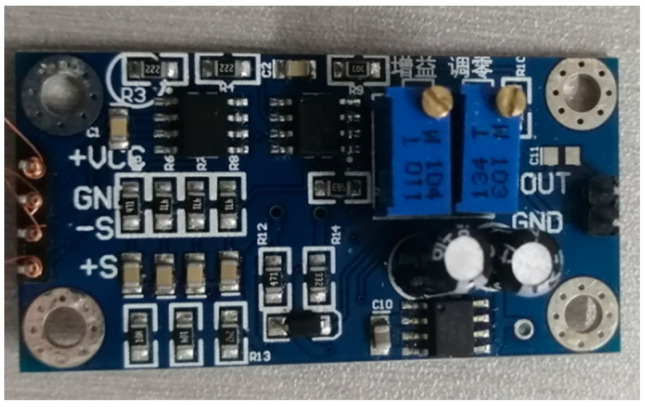
Circuit design of the flexible thermocouple film sensor.

**Figure 15 micromachines-13-01873-f015:**
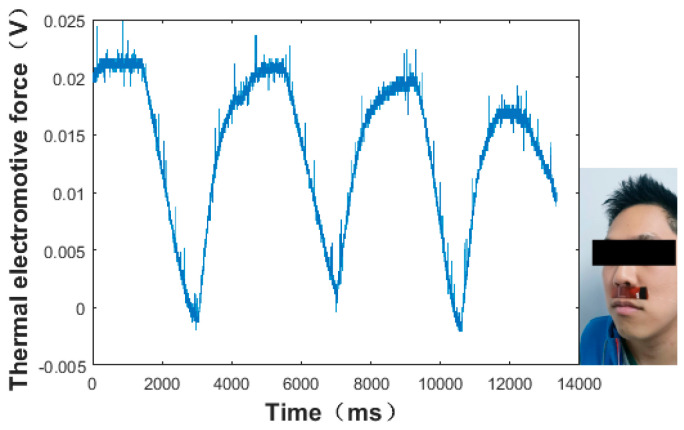
Test results of the normal respiration of the flexible thermocouple film sensor in a cycle.

**Figure 16 micromachines-13-01873-f016:**
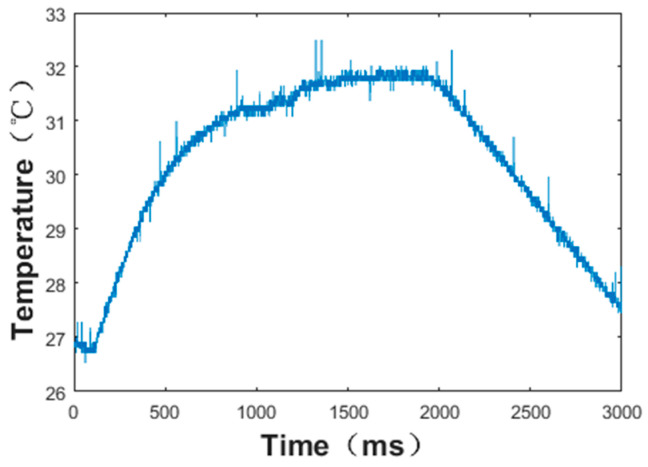
Temperature change of thermocouple hot junction under normal respiration conditions.

**Figure 17 micromachines-13-01873-f017:**
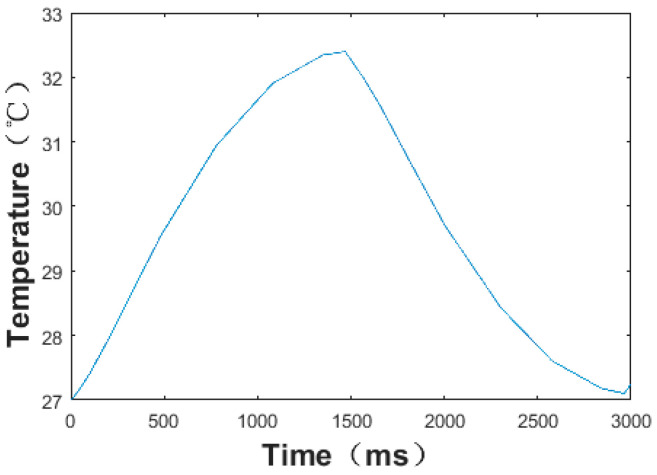
Simulation curve of thermocouple junction end temperature change under normal respiration conditions.

**Figure 18 micromachines-13-01873-f018:**
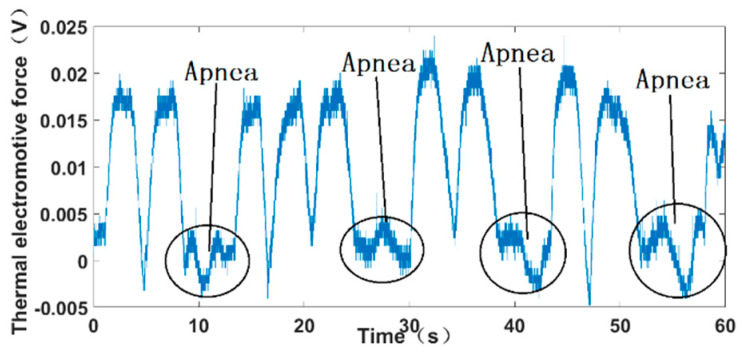
Test results under simulated apnea.

**Table 1 micromachines-13-01873-t001:** Response of the thermocouple with different thickness.

Thickness of Thermocouple (nm)	Response Time (ms)
500	40
750	57
1000	79
1250	104
1500	134

**Table 2 micromachines-13-01873-t002:** Average temperature difference of the hot junction under different respiratory flows.

Respiratory Flow (%)	Average Temperature Difference (°C)
100	5.17
90	4.70
80	4.10
70	3.50
60	3.21
50	2.45

**Table 3 micromachines-13-01873-t003:** Average temperature difference of the hot junction under different thin-film thicknesses.

Thin-Film Thickness (μm)	Average Temperature Difference (°C)
50	8.47
75	6.74
100	5.17
125	4.07
150	3.52

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
