# Peer review of "A Flexible Thermocouple Film Sensor for Respiratory Monitoring"

_micromachines, 2022, doi:10.3390/mi13111873_

Round 1
Reviewer 1 Report
The authors describe a thermocouple film sensor on a polyimide substrate for respiration monitoring. Although the work showed that the different between the Apnea and normal breathing can measured by the output voltage, in my opinion, the work is not good enough to be published in this journal at this time. The reviewer's comments are as follows:
1. The author citied some literatures in the manuscript. What are the special advantages of this sensor compared to the sensors in other literature? This may need to discuss in the introduction part.
2. The authors claim that “two isosceles-square parts was kept at 20℃”. From my point of view, this maybe not be the reasonable parameter setting. Based on the devices design in this manuscript, it only a short distance between the skin and two isosceles-square parts. If the ambient temperature was set as 20 ℃and the skin temperature was set as 36 ℃, the temperature of the two isosceles-square parts may should be higher than 20 ℃.
3. In the main part of the manuscript, authors need to rearrange the order of 3. Fabrication., it may not be suitable to put this part in here.
4. The author claim that “If the respiratory intensity decreases by 30%, it can be considered as hypopnea, and more than 50% is very serious hypopnea”. How could the author give this kind of verdict? Is there any citation to support this conclusion?
5. Please following the guideline of this journal, removing this “0. How to Use This Template” part; and modifying the following part such as “1.2.1Simulation of the polyimide film”, “Author Contributions”, “Funding” and “References”.
6. There are many typographical and grammatical errors in the manuscript that the authors must carefully correct.
Author Response
Thank you very much for your comments on my manuscript, and the response are as follows:
The authors describe a thermocouple film sensor on a polyimide substrate for respiration monitoring. Although the work showed that the different between the Apnea and normal breathing can measured by the output voltage, in my opinion, the work is not good enough to be published in this journal at this time. The reviewer's comments are as follows:
1.The author citied some literatures in the manuscript. What are the special advantages of this sensor compared to the sensors in other literature? This may need to discuss in the introduction part.
Answer:The thermocouple film sensor was fabricated on the polyimide layer and MEMS technology with three sputtering steps, which was simple to fabricate with low cost. As a result, it is easy and comfortable for wearing outside the hospital.
The corresponding revision is on page1 line 74-77.
2.The authors claim that “two isosceles-square parts was kept at 20℃”. From my point of view, this maybe not be the reasonable parameter setting. Based on the devices design in this manuscript, it only a short distance between the skin and two isosceles-square parts. If the ambient temperature was set as 20 ℃and the skin temperature was set as 36 ℃, the temperature of the two isosceles-square parts may should be higher than 20 ℃.
Answer:The two isosceles-square parts was suspended, since the flexible polymide substrate and it’s thermal conductivity is low. As a result, they could be kept at a constant temperature.In the simulation and measurement, the enviroment was set as 20 ℃. In the future research, the variation of the environment temperature would be studied.
3.In the main part of the manuscript, authors need to rearrange the order of 3. Fabrication., it may not be suitable to put this part in here.
Answer:The research of the sensor was consisted of the design,simulation,fabrication and measurment. The fabrication is following the simulation after the key parameters was design and optimized by the simulation as follows:
In the design and simulation mentioned above, the key parameters were optimized such as the distribution of the thermocouple, the thickness of the polyimide and linewidth of the thermocouple. The flexible thermocouple sensor is fabricated by microfabrication technology, which were mainly composed of the sputtering technology.
The corresponding revision is in line 250-253.
4.The author claim that “If the respiratory intensity decreases by 30%, it can be considered as hypopnea, and more than 50% is very serious hypopnea”. How could the author give this kind of verdict? Is there any citation to support this conclusion?
Answer:It is derived from the definition of the AHI, which defined as: (number of sleep apnea+number of hypoventilation)/total sleep time. The recommended definition of hypoventilation requires that the airflow decreases by more than 30% for more than 10 seconds, and the rate of decrease in blood oxygen saturation is greater than or equal to 4%. Another definition of hypoventilation requires a 50% or greater decrease in gas for 10 seconds or more, accompanied by a decrease in blood oxygen saturation of 3% or more. Different definitions of hypoventilation may lead to quite different AHI values.
5.Please following the guideline of this journal, removing this “0. How to Use This Template” part; and modifying the following part such as “1.2.1Simulation of the polyimide film”, “Author Contributions”, “Funding” and “References”.
Answer:The “0. How to Use This Template” part was removed. And the following part was corrected
The “Author Contributions”, “Funding” and “References” parts were completed as shown in line 369-377
- There are many typographical and grammatical errors in the manuscript that the authors must carefully correct.
Answer: The typographical and grammatical errors were carefully corrected.such as abstract in line 11-19.

Reviewer 2 Report
Author reported a novel flexible thermocouple film sensor formed on flexible PI substrate. Overall, the works were proceeded well and the results are interesting. However, this paper doesn't contain the experimental section. Although author described their procedure in the body text, more detailed information must be given as a separate section, i.e. experimental. The manuscript is needed to be re-organized.
Author Response
Thank you very much for your comments on my manuscript, and the response are as follows:
Author reported a novel flexible thermocouple film sensor formed on flexible PI substrate. Overall, the works were proceeded well and the results are interesting. However, this paper doesn't contain the experimental section. Although author described their procedure in the body text, more detailed information must be given as a separate section, i.e. experimental. The manuscript is needed to be re-organized.
Answer: The experiments of the signal measurement system and process were completed as follows.
The measurement system was established as shown in Fig.13. The output of the sensor was connected with the oscilligraphs and an amplifier with a power supply of 1.5Vx4 .In the measurement system, the thermocouple signal is weak and need to be amplified. A kind of the low-power, high-precision amplifier AD620 was applied, which can set the magnification from 1 to 1000 times with only one external resistor. The maximum input offset voltage is 50 μv. The circuit diagram and the print circuit board(PCB) is shown in Fig. 14.The ambient temperature was 27 ℃, and the multiple of the amplifier was set to 100 times.
The corresponding revision is in line292-299.
Then the the sensor was attached to the bottom of the nose, which the input part was inside the nose and the output part was outside the nose as shown in Fig15. The different respiratory signal of the sensor was tested by the oscilloscope, when the normal and abnormal respiration was modified.The measurement result were obtained as shown in Fig.15-17.
The corresponding revision is in line 306-310.

Reviewer 3 Report
This well-written manuscript presents interesting results on the use of microfabricated thermocouples to measure respiratory signals that can indicate sleep apnea. The design of the microfabricated device is well documented, and the simulations are relevant. However, the following questions need to be clarified:
1. There are many kinds of airflow sensors for detecting respiratory flow (DOI: 10.1109/JSEN.2022.3167023, https://doi.org/10.1016/j.bios.2020.112288), so why the thermocouple is used for respiration detection. What is the advantage of using this technology? The comparison with other sensors needs to be analyzed.
2. The temperature difference between hot and cold junctions is important for respiration sensing. The paper mentioned that “In order to ensure the temperature difference between the hot and cold junction, the cold junction of the thermocouple is leading out to keep unchanged and avoid effecting by the skin.” Please provide more details on how to realize the thermal isolation for the device.
Author Response
Thank you very much for your comments on my manuscript, and the response are as follows:
This well-written manuscript presents interesting results concerning the use of microfabricated thermocouples to measure respiratory signals that can indicate sleep apnea. The design of the microfabricated device is well documented, and the simulations are relevant. The following questions :
- There is kinds of airflow sensors for detecting respiratory flow(DOI: 1109/JSEN.2022.3167023, https://doi.org/10.1016/j.bios.2020.112288), why using thermocouple to detect respiration. What is the advantage using this sensor? The comparison with other sensors need to be analyzed.
Answer: The research other kinds of airfolw sensors was cited as reference [24]and [25]. The advantage of the flexible thermalcouple sensor was described as follows :This paper presented a flexible thermocouple film sensor, which was simple to fabricate with low cost, since it was fabricated on the polyimide layer and MEMS technology with three sputtering steps. As a result, it is easy and comfortable for wearing outside the hospital.
The corresponding revision is in line 74-77
- The temperature difference between hot and cold junctions is important for respiration sensing. The paper mentioned that “In order to ensure the temperature difference between the hot and cold junction, the cold junction of the thermocouple is leading out to keep unchanged and avoid effecting by the skin.” Please provide more details how to realize the thermal isolation for the device?
Answer:The two isosceles-square parts was suspended, since the flexible polymide substrate and it’s thermal conductivity is low. As a result, they could be kept at a constant temperature.

Round 2
Reviewer 1 Report
1.The author citied some literatures in the manuscript. What are the special advantages of this sensor compared to the sensors in other literature? This may need to discuss in the introduction part.
Answer:The thermocouple film sensor was fabricated on the polyimide layer and MEMS technology with three sputtering steps, which was simple to fabricate with low cost. As a result, it is easy and comfortable for wearing outside the hospital.
R1: The authors may still need to add some discussion for comparing with other literature, not only just presenting the advantages of this device.
4.The author claim that “If the respiratory intensity decreases by 30%, it can be considered as hypopnea, and more than 50% is very serious hypopnea”. How could the author give this kind of verdict? Is there any citation to support this conclusion?
Answer:It is derived from the definition of the AHI, which defined as: (number of sleep apnea+number of hypoventilation)/total sleep time. The recommended definition of hypoventilation requires that the airflow decreases by more than 30% for more than 10 seconds, and the rate of decrease in blood oxygen saturation is greater than or equal to 4%. Another definition of hypoventilation requires a 50% or greater decrease in gas for 10 seconds or more, accompanied by a decrease in blood oxygen saturation of 3% or more. Different definitions of hypoventilation may lead to quite different AHI values.
R4: For this part, it may still be needed to cite some literature to support your discussion and revised the manuscript.
Author Response
Thank you very much for your comments, and the response are as follows:
R1: The authors may still need to add some discussion for comparing with other literature, not only just presenting the advantages of this device.
Answer: Compared with the work in reference [22][23], which the air flow sensor was fabricated based on silicon substrate and need extra constant power supply, this paper presented a thermocouple thin film flow sensor, which is fabricated based on polymide substrate with three sputtering steps based on the thermal electromotive force without extra power supply. As a result, it is more flexible for wearable diagnose system outside the hospital with lower cost.
The corresponding is in line 66-71
R4: For this part, it may still be needed to cite some literature to support your discussion and revised the manuscript.
The reference [3-6]was cited as in line 209-211.

Reviewer 2 Report
1. There are many typos and grammatical errors. e.g. Start of introduction, it should indicate the full name of OSAHS even though they mentioned in abstract. There were many more examples. Authors suggest to carefully check all the sentences.
2. Since the paper proposed "flexible sensor", I would suggest to include some comments on the reliability on the flexible electrodes in the introduction with the proper references such as (http://casopisi.junis.ni.ac.rs/index.php/FUMechEng/article/view/10949).
Author Response
Thank you very much for your comments, and the response are as follows:
- There are many typos and grammatical errors. e.g. Start of introduction, it should indicate the full name of OSAHS even though they mentioned in abstract. There were many more examples. Authors suggest to carefully check all the sentences.
Answer: The OSAHS was corrected as Obstructive Sleep Apnea-hypopnea Syndrome in the introduction as in line23. And the other senctences was corrected.
- Since the paper proposed "flexible sensor", I would suggest to include some comments on the reliability on the flexible electrodes in the introduction with the proper references such as (http://casopisi.junis.ni.ac.rs/index.php/FUMechEng/article/view/10949).
Answer: The reliable of the flexible electrodes would be further studied and the reference was cited as reference [26] as in line 415.

Round 3
Reviewer 2 Report
There are still typos and grammatical errors. I suggest correct the errors during proof reading process.